

# Single-particle spectra and magnetic susceptibility in the Emery model: A dynamical mean-field perspective

Yi-Ting Tseng[1], Mário O. Malcolms[2], Henri Menke[1],
Marcel Klett[2], Thomas Schäfer[2] and Philipp Hansmann[1]⋆

**1** Department of Physics, Friedrich-Alexander-Universität
Erlangen-Nürnberg, 91058 Erlangen, Germany
**2** Max Planck Institute for Solid State Research,
Heisenbergstraße 1, 70569 Stuttgart, Germany

⋆ philipp.hansmann@fau.de

## Abstract

We investigate dynamical mean-field calculations of the three-band Emery model at the one- and two-particle level for material-realistic parameters of high-$T_c$ superconductors. Our study shows that even within dynamical mean-field theory, which accounts solely for temporal fluctuations, the intrinsic multi-orbital nature of the Emery model introduces effective non-local correlations. These correlations lead to a non-Curie-like temperature dependence of the magnetic susceptibility, consistent with nuclear magnetic resonance experiments in the pseudogap regime. By analyzing the temperature dependence of the uniform static spin susceptibility obtained by single-site and cluster dynamical mean-field theory, we find indications of emerging oxygen-copper singlet fluctuations, explicitly captured by the model. Despite correctly describing the hallmark of the pseudogap at the two-particle level, such as the drop in the Knight shift of nuclear magnetic resonance, dynamical mean-field theory fails to capture the spectral properties of the pseudogap.

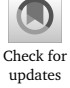

# 1  Introduction

Predictive calculations for emergent phenomena in correlated condensed matter systems are a contemporary focus of a large research community. One of the principal and notorious paradigms in this field is the unconventional superconductivity in hole- and electron-doped cuprates [1, 2]. The ionic electronic configuration of copper in these compounds suggests a single half-filled band of $d_{x^2-y^2}$ character which is usually modeled by the famous single-band Hubbard model [3–6]. Alternative models that include also oxygen $2p$-orbitals were proposed by Emery [7] and refined by Andersen [8]. While the single-band model is, due to its simplicity, very attractive for the treatment even with demanding quantum many-body methods [9, 10], the appeal of the Emery model is the explicit inclusion of higher energy excitations such as charge-transfer processes.

   To reach superconductivity, cuprates must be doped at sufficiently low temperatures [11, 12]. Therefore, the accuracy of a calculation for cuprates is primarily evaluated by comparison to experiments in the temperature-versus-doping phase diagram. From the theoretical perspective, the ideal calculation should allow for predictions of all trends in this phase diagram instead of capturing single points with adjusted parameters. At zero doping and sufficiently high temperature, the paramagnetic insulating state is captured by dynamical mean-field theory (DMFT) [13–15] as a Mott insulator. On the other side of the doping axis, at extremely high hole concentrations, the system behaves like a normal Fermi liquid, which can also be captured by DMFT. The challenge lies in connecting these extremes and capturing the famous pseudogap phase [16–18], as well as the superconducting dome below a strange metallic phase that does not behave like a Fermi liquid (e.g., optical conductivity [19]) and has a non-Curie-like magnetic response [11]. Nuclear magnetic resonance (NMR) has proven to be an indispensable experimental tool for analyzing magnetic properties in these regimes [16, 20–22], specifically for the onset of the pseudogap, which is signaled by a drop in the NMR Knight shift.

   The strength of DMFT is its ability to cross non-perturbatively from the weak to the strong coupling limit. Yet, due to the local nature (i.e., no momentum dependence of correlations) of the DMFT approximation its application to the two-dimensional single-band Hubbard model fails dramatically when compared to experiments at low temperatures. Curing this deficiency of DMFT towards the inclusion of non-local correlations became the motor for the development of a plethora of cluster-based or diagrammatic extensions of DMFT [23, 24]. Instead, when DMFT is applied to the three-band model in the localized Wannier orbital basis, its self-energy in the basis of the quasiparticle bands becomes momentum-dependent due to the mixed orbital nature of the Bloch bands. Such effective non-locality within the DMFT framework is known and has been pointed out, e.g. in multi-orbital models for nickelates [25, 26], and ruthenates [27, 28]. While much fewer in number compared to the single-band case, there have been some studies of the Emery model with DMFT showing good agreement with experiments at the one-particle level [29–32].

   In the present work, we focus on the application of single-site DMFT to the Emery model, emphasizing doping and temperature dependencies at the single- and two-particle levels. Our results include single-particle spectra and static spin susceptibilities as a function of temperature and doping. We find good agreement of our calculations with nuclear magnetic resonance

(NMR) experiments of Sr-doped $La_{2-x}Sr_xCuO_4$ [21,33,34]. Our results stress the fact that the drop in the spin susceptibility as a function of the temperature can be explained by a oxygen-copper singlet formation and occurs without the opening of a momentum-selective gap in the single-particle spectra (in contrast to a bubble diagram RPA-like interpretation for which such a gap would be required).

## 2 Models and method

In the present study, we analyze the three-band Emery model within DMFT using material-realistic ab-initio model parameters (hopping and interaction) derived in [35, 36]. Specifically, the three-band model consists of one planar Cu $d_{x^2-y^2}$ and two oxygen ($p_x$ and $p_y$) basis orbitals in the unit cell. While the original formulation [7] included only one $d$-$p$ hopping integral in addition to the Hubbard interaction on the $d_{x^2-y^2}$ orbital, later works, particularly by Andersen et al. [8], refined the parameters to be closer to a material-realistic regime, including intra-oxygen $t_{pp}$ terms. For our model, we use the tight-binding hopping and interaction parameters established in previous works on the Emery model [35,36]. The Hamiltonian reads

$$H_{dp}(\mathbf{k}) = H_0^{dp}(\mathbf{k}) + H_{int.}^{dp}(\mathbf{k}), \tag{1}$$

where $H_0^{dp}(\mathbf{k})$ is the single-particle hopping part

$$H_0^{dp}(\mathbf{k}) = \begin{pmatrix} \varepsilon_d & t_{pd}(1-e^{-ik_x}) & t_{pd}(1-e^{-ik_y}) \\ t_{pd}(1-e^{ik_x}) & \varepsilon_p + 2t'_{pp}\cos k_x & t_{pp}(1-e^{ik_x})(1-e^{-ik_y}) \\ t_{pd}(1-e^{ik_y}) & t_{pp}(1-e^{-ik_x})(1-e^{ik_y}) & \varepsilon_p + 2t'_{pp}\cos k_y \end{pmatrix}. \tag{2}$$

Here $\varepsilon_d$ and $\varepsilon_p$ are the on-site energies of $d$- and $p$- orbitals. $t_{pd}$ is the nearest-neighbor hopping between Cu and O, $t_{pp}(t'_{pp})$ is the (second) nearest-neighbor hopping between oxygen orbitals as shown in schematic Fig. 1. Following Refs. [35,36] we use $\varepsilon_d = 0.0$ eV, $\epsilon_p = 1.50$ eV, $t_{pd} = 1.37$ eV, $t_{pp} = 0.65$ eV and $t'_{pp} = 0.13$ eV (note that the onsite energies include already an atomic-limit double-counting corrections). The corresponding non-interacting bandstructure is shown on the right-hand side of Fig. 1 where we used the orbitally resolved $\mathbf{k}$-dependent eigenvectors of $H_0^{dp}(\mathbf{k})$ for the color coding of the bands.

For the interacting part of the Hamiltonian (1) we follow [36] and include only $U_{dd} = 9.1$eV as a local Hubbard repulsion

$$H_{int.}^{dp}(\mathbf{k}) = U_{dd}\hat{n}_{d\uparrow}\hat{n}_{d\downarrow}, \tag{3}$$

where $n_{d\sigma}$ is the density operator of quasiparticles in the Cu-$d$. As [36] we neglect $U_{pp}$ and $U_{pd}$ beyond their effect on the static mean-field level which is already included in the effective single-particle energies $\varepsilon_d$ and $\varepsilon_p$.

**DMFT and CDMFT solution.** To solve the model, we perform (single-site and cluster) DMFT calculations using continuous-time quantum Monte Carlo impurity solvers [37] in the hybridization expansion (CT-HYB). In DMFT the interacting lattice problem is solved by mapping it to an auxiliary single impurity Anderson model [13–15]. While such a mapping is exact in the limit of infinite coordination number, in finite dimensions it is an approximation that accounts for fully dynamic but purely local correlations on the level of the single-particle self-energy. For this study we use the well established w2dynamics code [38] and consider the $d_{x^2-y^2}$ orbital as the impurity in our $dp$-model with a local Hubbard term of $U_{dd} = 9.1$ eV. For the computation of the magnetic susceptibility (see section 3.2), we performed single-site

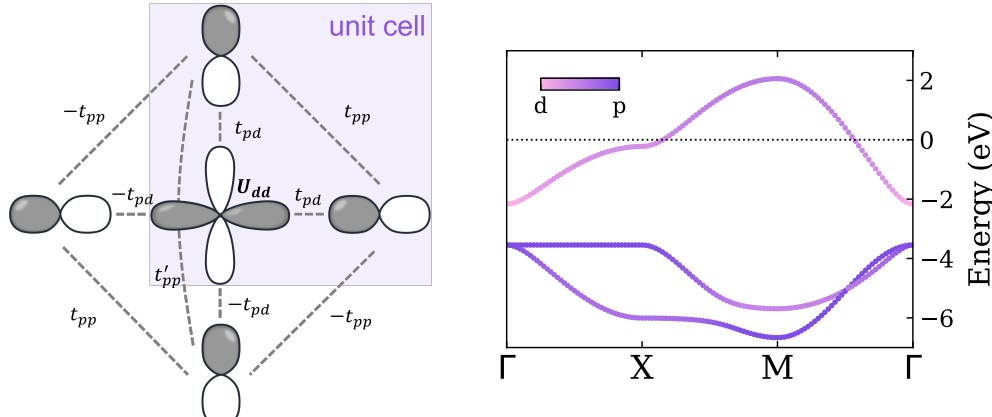

Figure 1: Left-hand side: Unit cell, hopping, and interaction parameters in the Emery model. Right-hand side: Non-interacting band structure with color coded copper d- (pink) and oxygen p- (dark violet) orbital character.

DMFT and $2 \times 1$ CDMFT calculations [23,39]. The latter allows us to explicitly include nearest-neighbor Cu-Cu correlations and to test the validity of the single-site approximation for the susceptibility in the Emery model.

**Analytic continuation of the self-energy.** To obtain the real-frequency self-energy, we perform a maximum entropy (MaxEnt) analytic continuation on an auxiliary Green function

$$G^{\text{aux}}(i\omega_n) = (i\omega_n + \mu^{\text{aux}} - \Sigma_{dd}(i\omega_n))^{-1}, \tag{4}$$

with the $\Omega$MaxEnt code [40] and subsequent inversion of the equation

$$\Sigma_{dd}(\omega) = \omega + \mu^{\text{aux}} - G^{\text{aux}}(\omega)^{-1}, \tag{5}$$

which we can use to compute the retarded lattice Green function

$$G_{\text{lat}}(\omega, \mathbf{k}) = \left((\omega + \mu + i\delta)\mathbf{1} - H_0^{dp}(\mathbf{k}) - \Sigma^{\text{DMFT}}(\omega)\right)^{-1}, \tag{6}$$

where the self-energy $\Sigma^{\text{DMFT}}(\omega) = \text{diag}(\Sigma_{dd}(\omega), 0, 0)$ is a diagonal $3 \times 3$ matrix with only one finite element, i.e. $\Sigma_{dd}(\omega)$ in the $d$-subspace. As the exact result of the self-energy is independent of $\mu^{\text{aux}}$ in Eq. (4), we used variations of this parameter to estimate the stability of the analytical continuation by means of error bars for the spectral function (gray shaded areas in Figs. 2 and 3). Additionally, the analytical continuation of the level of the self-energy has the advantage to avoid artificial broadening effects which results in a much higher resolution of the spectra.

**Uniform magnetic susceptibility.** In order to extract the uniform/ferromagnetic, i.e. $\mathbf{q} = \mathbf{0}$, spin susceptibility, we measure the magnetization, $m = n_{d\uparrow} - n_{d\downarrow}$, in the presence of a small ferromagnetic field within the linear response regime, with field strength $H = 6.5$ meV, and we obtain the slope of a linear fit which enables us to compute the uniform susceptibility in the static limit as

$$\chi_{dd}(\mathbf{q} = \mathbf{0}) = \left.\frac{\partial m}{\partial H}\right|_{H=0}. \tag{7}$$

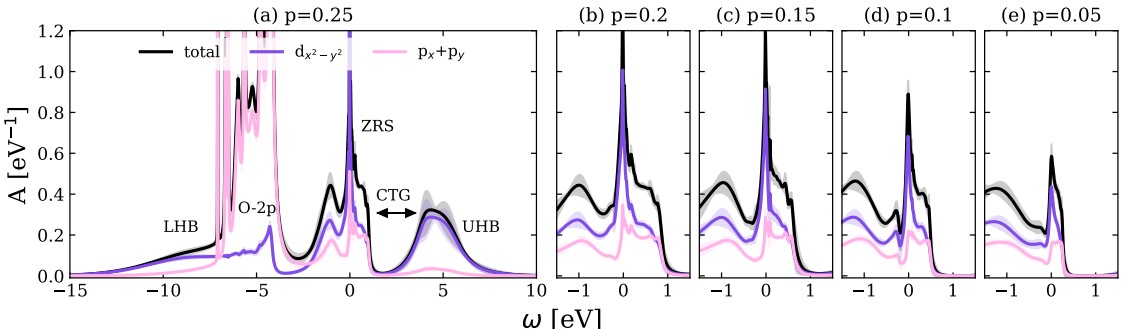

Figure 2: Doping- and orbital-dependent local spectral function $A_\alpha(\omega) = -\frac{1}{\pi}\sum_{\mathbf{k}}\text{Im}[G_{\text{lat}}(\omega,\mathbf{k})]_{\alpha,\alpha}$ for $U_{dd} = 9.1\,\text{eV}$ and $T \approx 150\,\text{K}$. The (light) black, purple, and pink line shows spectral function (error bar) contributed by total, $d_{x^2-y^2}$ and $p_x + p_y$ orbitals. The main features are indicated by labels and include the lower (LHB) and upper (LHU) Hubbard bands, $p$-bands which define a charge transfer gap (CTG), and a low energy Zhang-Rice singlet (ZRS) quasiparticle peak at the Fermi energy.

Experimentally, the static susceptibility can determined from the Knight shift measured by nuclear magnetic resonance (NMR) [41] as the two are linearly related by the hyperfine coupling constant [42, 43]. Hence, we can compare theory to experiment by a scatter plot of the calculated susceptibility for different doping and temperatures versus the experimentally measured Knight shift which - ideally - results in a straight line with the coupling constant as a slope. These plots are commonly referred to as Clogston–Jaccarino (CJ) plots.

## 3  Results

### 3.1  Single-particle spectra

In Fig. 2 we show the **k**-integrated orbitally resolved spectral function for the Emery model defined in Eq. (1):

$$A_\alpha(\omega) = -\frac{1}{\pi}\sum_{\mathbf{k}}\text{Im}[G_{\text{lat}}(\omega,\mathbf{k})]_{\alpha,\alpha}\,, \tag{8}$$

where $\alpha$ is the orbital index. The computations have been performed for $U_{dd} = 9.1\,\text{eV}$ and at the temperature of $T \approx 150\,\text{K}$. In the panels from left to right we show results for different doping levels ranging from $p = 0.25$ to $p = 0.05$, where the doping level is defined as $p \equiv 5.0 - N$ (i.e. $p = 0.0$ corresponds to "half-filling"). In the plots we resolve the spectra into copper $d$- (dark violet) and oxygen $p$- (pink) orbital characters respectively. The shaded areas around the lines correspond to an error estimate (see Sec. 2 for details).

In the leftmost panel we show the spectra for the $p = 0.25$ case in a wide energy range from $-15\,\text{eV}$ to $+10\,\text{eV}$ and label the most significant spectral features. At larger energies we see clear Hubbard bands of copper-$d$ character which are centered around $-10\,\text{eV}$ and $+5\,\text{eV}$. In the area between $-3\,\text{eV}$ and $-8\,\text{eV}$ most of the oxygen-$p$ spectral weight is found which in large parts is unaffected by correlation effects and identical to the DFT single-particle density of states (DOS) of oxygen (note that as we perform analytical continuation on the level of the self-energy which is finite only for the Cu d-orbital, the oxygen peaks are not subject to life time- and/or spurious MaxEnt broadening and remain maximally sharp). The most intriguing feature, however, is the sharp and pronounced quasiparticle peak at the Fermi level

$\omega = 0$ of strongly mixed copper-$d$ and oxygen-$p$ character. If we follow this feature towards the half-filled limit ($p = 0.0$) we observe the expected sharpening and decrease in spectral weight which eventually will result in the metal-to-insulator transition and the opening of a gap at $p = 0.0$. The mixed copper-oxygen nature of this quasiparticle has been attributed to the formation of a Zhang-Rice singlet already in previous works [29–32, 36, 44]. We note that the charge-transfer gap (CTG) of our Emery-spectra is roughly between $1.5 - 2\text{eV}$ for lower doping and increases for larger values of $p$. This energy scale (and indeed the doping trend) is in agreement with experiments on the optical conductivity for Sr doped La$_2$CuO$_4$ [45].

Next, we analyze the analytically continued DMFT self-energy in the orbital basis and plot it in panel (a) of Fig. 3 covering the same doping levels as for the spectra. Also here the shaded areas indicate error bars estimated for the analytical continuation. At the highest doping levels the self-energy shows pronounced Fermi liquid behaviour for $T \approx 150\,\text{K}$ with a linear real part (from which the correlation induced mass-renormalization can be estimated) and an imaginary part $\propto \omega^2$. Closer to half-filling the real-part remains linear, but the quasiparticle scattering rate (i.e. $-\text{Im}\,\Sigma(\omega = 0)$) increases significantly with decreased hole-doping at constant temperature. Unsurprisingly, the **k**-independent DMFT self-energy fails to capture the **k**-selective pseudogap physics at small doping values. As we mentioned above, a projection on the quasiparticle band around $\varepsilon_F$ would result in an effective **k**-dependence of the self-energy for the low-energy quasiparticles as it would also implicitly encode information about the **k**-dependent composition of the quasiparticles in terms of copper-$d$ and oxygen-$p$ character [25–28]. While such projection results in a **k**-dependent mass renormalization, it would not change the overall Fermi-liquid behaviour and therefore cannot cure the DMFT shortcomings in capturing pseudogap physics, i.e., the gapped behaviour of the single-particle spectrum in the anti-nodal region.

We can, however, investigate whether the dependence of **k**$_F$ on the $k_{Fx} = k_{Fy}$ diagonal, is captured correctly by our calculations. To check this we plot the position of the quasiparticle pole on that diagonal for different dopings in panel (b) of Fig. 3 [same color coding as in panels (a)] and compare it to the experimental data shown in grey. As we can see the position of the nodal quasiparticle **k**$_F$ in the Brillouin zone is captured well by the DMFT spectral function for the Emery model: in our calculations as well as in the experiment for increased hole doping the nodal point is dragged towards the $\Gamma$-point of the Brillouin zone. For comparison we provide the result for the corresponding single-band calculation in the appendix App. A.

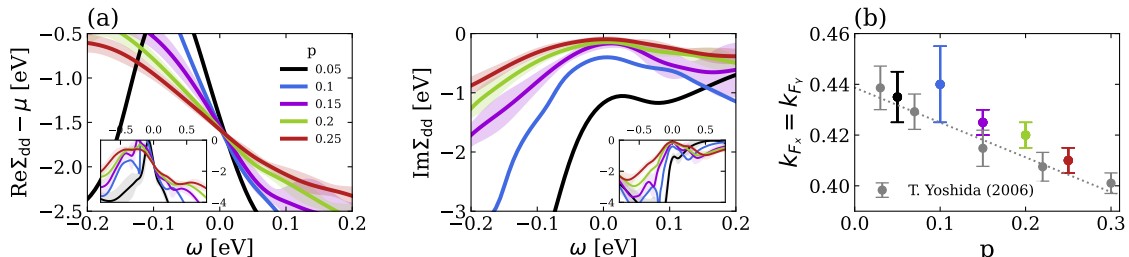

Figure 3: (a) Doping-dependent real-frequency self-energy (real and imaginary parts) of the $d$-orbital from analytic continuation. The linear behavior of $\text{Re}[\Sigma(\omega)] \propto \omega$ and $\text{Im}[\Sigma(\omega)] \propto \omega^2$ around $\omega = 0$ indicate clearly a Fermi-liquid behavior. (b) Doping dependence of the $k_F$ value in the nodal direction. The calculation has been performed at $T \approx 150$ K. Experimental data are adapted from angle-resolved photoemission spectroscopy (ARPES) [46].

## 3.2 Magnetic susceptibility

We now turn to the results for the static uniform magnetic susceptibility. Fig. 4 (left) shows $\chi_{dd}(\mathbf{q} = \mathbf{0})$. In order to distinguish the effect of copper-oxygen correlations (included in single-site DMFT) from non-local copper-copper correlations, we performed $2 \times 1$-cluster DMFT calculations. Our data covers temperatures from 250 K down to 30 K and the same doping ranges ($p = 0.05$ to $0.24$) as for the single-particle spectrum.

For both single-site and cluster DMFT calculations, we observe a monotonous increase of $\chi_{dd}(\mathbf{q} = \mathbf{0})$ upon hole-doping when keeping the temperature fixed. Further, for fixed doping, the temperature dependence has non-Curie-like behavior (i.e., a downturn at lower temperatures) for all considered doping levels and shows a maximum that moves from $T_{\max} \approx 130$ K for the lowest doping to $T_{\max} \approx 100$ K at maximum doping. The observed behavior seems to be in line with experimental measurements of the NMR Knight shift for cuprates [21,33,34,47–49]. For a more quantitative comparison, we show the Clogston-Jaccarino plot in comparison to LSCO data in the right panel of Fig. 4. The points in this scatter plot have the experimentally observed Knight shift for $La_{2-x}Sr_xCuO_4$ for given temperature and doping as $x$-coordinate and the DMFT calculated $\chi_{dd}$ for the same parameters as $y$-coordinate. The plot confirms a linear relation between the two quantities and, therefore, a very good agreement of the temperature and doping trends in $\chi_{dd}$ captured by the calculation with the NMR measurement. In App. B we show that the occurrence of the temperature maximum of $\chi_{dd}$ in DMFT is indicative of the emergence of copper-oxygen (Zhang-Rice) singlet fluctuations, which, for the Emery model, are included even in a single-site DMFT calculation. This claim is further supported by comparison to the cluster DMFT calculations (open triangle symbols in Fig. 4), which demonstrate that explicit inclusion of nearest neighbor Cu-Cu correlations do not alter the temperature or doping trend we found. This observation is a strong indication that Cu-O correlation and the formation of Cu-O singlet fluctuations are key to understand the experimentally observed behavior. We note in passing that our results agree well with recent studies on the periodic Anderson model [50], which, on a technical level, is very similar to the Emery model. Notably, the doping trend of $\chi_{\mathrm{imp}}$ is inverted compared to the uniform $\chi_{dd}(\mathbf{q} = \mathbf{0})$ (see Fig. 7 in Appendix C) which was also reported in [50]. This signals the departure of atomically localized moments on the copper site upon doping and highlights the increasing importance of fluctuations around the Fermi level in determining the *uniform* magnetic response. As for the single-particle spectrum, we provide the corresponding results for the single-band model in the appendix A.

At a first glance, it seems odd that single-site DMFT for the Emery model reproduces the two-particle observable $\chi_{dd}$ better than the single-particle spectrum, for which it fails to capture the correct pseudogap behavior of the single-particle spectral function at lower doping levels. On the one hand it might be that the static *uniform* magnetic susceptibility measured by the NMR Knight shift is simply not too sensitive with respect to non-local correlations which are responsible for momentum-selective gaping of the Fermi surface. On the other hand there might be a methodological reason as well: In single-site DMFT, the correlations which are responsible for the downturn of $\chi_{dd}(\mathbf{q} = \mathbf{0})$ at lower temperatures give no feedback to the single-particle level. In order to test if a feedback of the two-particle level improves the single-particle spectra of the Emery model, calculations with methods incorporating these non-local correlations in the single-particle quantities [e.g., the dynamical vertex approximation (DΓA), [24,51,52]] will be carried out in a follow-up study.

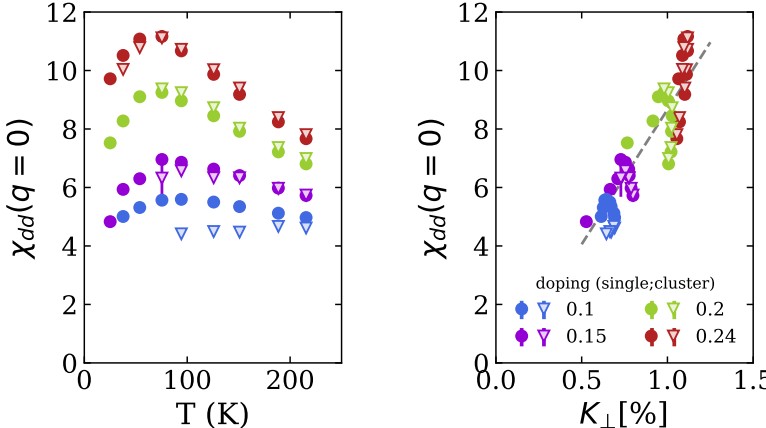

Figure 4: (left) dp-model uniform spin susceptibility on d-orbital with doping p=0.1-0.24 computed in single-site (filled circles) and $2 \times 1$ cluster- DMFT. (right) Clogston–Jaccarino plot of the calculated spin susceptibility (y-axis) versus the experimental Knight shift data (x-axis) from [21,33].

## 4 Conclusions

In summary we have revisited a material realistic Emery model for cuprate high-$T_c$ super-conductors. We analyzed the model within the DMFT framework and computed observables corresponding to the single-particle spectral function accessible in (AR)PES experiments, as well as the two-particle correlation function of the magnetic susceptibility related to the NMR Knight shift. Comparison to experiments of both observables revealed that their doping and temperature dependencies can be captured well in the Emery picture for both single- and two-particle quantities within *one single set of model parameters*. This observation in combination with benchmarks of the single-band model (especially for the magnetic susceptibility; see App. A) for fixed parameters suggests that single-site DMFT performs significantly better in predicting experimental observables for cuprates, and likely other charge transfer systems, when the oxygen degrees of freedom are included explicitly. Additional CDMFT calculations for the susceptibility on $2 \times 1$ supercells further indicates the important role of copper-oxygen correlations and the formation of Zhang-Rice singlets are key to interpret the experimentally observed downturn of the susceptibility at lower temperatures. This is in contrast to compounds which do not belong to the charge-transfer family like early transition metal oxides (TMO) like titanates and vanadates or those where the transition metal ocurs in a low valence state like in infinite-layer nickelates. Here, oxygen is believed to play a less crucial role and single-band calculations without explicit oxygen contributions yield reasonable agreement with doping trends observed in experiments [51,53–55]. Nevertheless, even in the nickelates, the question of the "minimal model" remains a topic of ongoing debate. We stress, however, that there is no doubt about the failure of single-site DMFT for low dimensional systems at low temperatures and **k**-selective gaps in the spectral function of the pseudogap regime remain surely beyond the grasp of an on-site self-energy even in the Emery model.

In follow-up studies we will continue our study for the Emery model beyond the single-site DMFT approximation. The inclusion of non-local correlations, e.g., with DΓA, on the two-particle level will allow for a description of single- and two-particle level on equal footing within the pseudogap regime of cuprate compounds [52].

## Acknowledgments

We thank Karsten Held, Matthias Hepting, Marc-Henri Julien, Giorgio Sangiovanni, Eric Jacob and Jörg Schmalian for helpful discussions.

**Funding information**   We acknowledge financial support by the DFG project HA7277/3-1. We thank the computing service facility of the MPI-FKF for their support and gratefully acknowledge the HPC resources (TinyFAT) administered by the RRZE of the Friedrich-Alexander-Universität Erlangen-Nürnberg (FAU).

## A   The single-band model

**Single-band model benchmarks**   The downfolding step from the three- [Eq. (1)] to the single-band model is trivial as a simple diagonalization of the hopping matrix in momentum space automatically projects out the well-separated band at the Fermi level:

$$H_0^d(k) = 2\tilde{t}_{dd}(\cos k_x + \cos k_y) + 4\tilde{t}'_{dd}\cos k_x \cos k_y + \dots \tag{A.1}$$

For our calculations, we use the numerically evaluated Hamiltonian directly in momentum space. The corresponding hopping parameters are $\tilde{t}_{dd} = -0.51\,\text{eV}$ for the nearest- and $\tilde{t}'_{dd} = 0.026\,\text{eV}$ for the next-nearest neighbor terms. The interaction $\tilde{U}_{dd}$ for the benchmark $d$-model calculations was adjusted by fitting the self-energy induced mass renormalization to the results of the $dp$-model. This estimate yielded $\tilde{U}_{dd} = 5\,\text{eV}$ for the corresponding single-band model.

Single-particle properties of $dp$-model can be well captured by the $d$-model as shown in Fig. 5 (a) and (b). The $d$-model shows Fermi-liquid-like self-energy and can reproduce the mass renormalization and $k_F$ at the nodal point with $\tilde{U}_{dd} = 5\,\text{eV}$. However, it fails on two-particle properties as Clogston–Jaccarino plot diverges [Fig. 5 (c)].

## B   The asymmetric dimer model

In order to aid with the interpretation of our main results for the temperature and doping dependence of the magnetic susceptibility and orbital occupation in the Emery model calculated with single-site DMFT, we consider a minimal toy model by replacing the bath with a single ligand, which can reproduce the simultaneous changes of d-occupation and susceptibility with temperature. The small cluster as a minimal toy model can be "derived" from a single isolated unit-cell of the Emery model. Within the unit-cell we can rotate the oxygen basis to a bonding and a (fully occupied and decoupled) non-bonding ligand state. The bonding ligand together with Cu $d_{x^2-y^2}$ forms an *asymmetric* Hubbard dimer which we can solve analytically (the results for ground and first excited state can be found in Tab. 1).

The Hamiltonian reads

$$\hat{H} = \sum_\sigma \left(\varepsilon_l \hat{n}_{l,\sigma} + t_{dl}(\hat{c}_{d\sigma}^\dagger \hat{c}_{l\sigma} + \hat{c}_{l\sigma}^\dagger \hat{c}_{d\sigma})\right) + U_{dd}\hat{n}_{d\uparrow}\hat{n}_{d\downarrow}, \tag{B.1}$$

with $t_{dl} = 1.93\,\text{eV}$, $\varepsilon_l - \varepsilon_d = 2.15\,\text{eV}$ and the onsite interaction of the $d_{x^2-y^2}$ orbital $U_{dd} = 9.1\,\text{eV}$. To approach the doped cases, we analyze the dimer at half-filling which corresponds to a filling of $N = 4$ in the Emery model and, therefore, represents the high doping limit. In Fig. 6 we show the resulting plots of the susceptibility together with temperature-dependent occupations of the $d_{x^2-y^2}$ orbital in the dimer-model as well as the corresponding DMFT plots at the highest considered doping level.

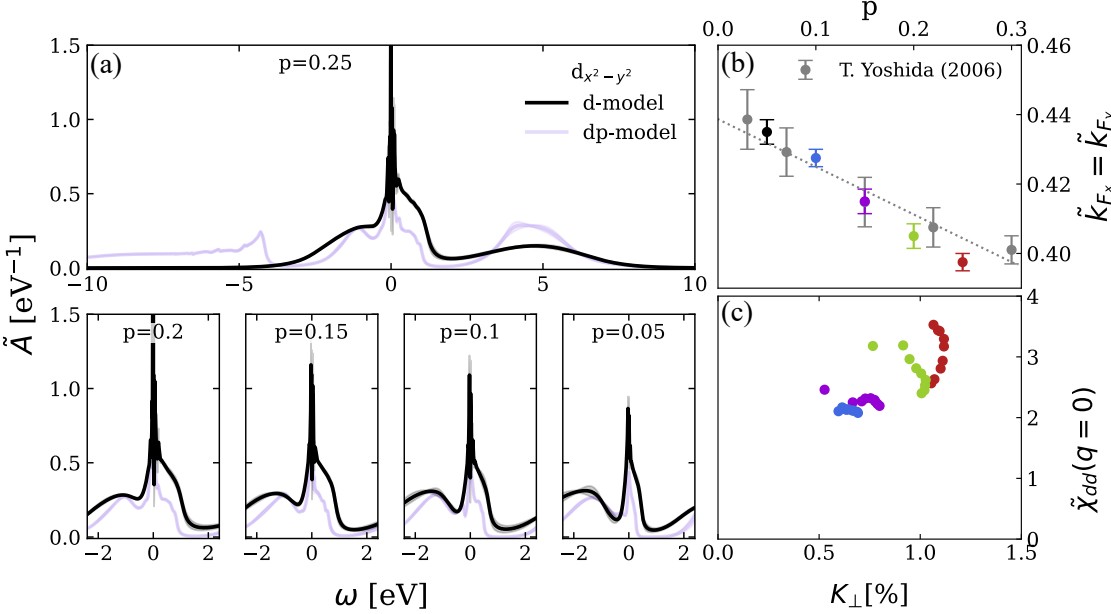

Figure 5: Single-band model benchmarks. (a): Doping- and orbital-dependent local spectral function $\tilde{A}_\alpha(\omega) = -\frac{1}{\pi}\sum_{\mathbf{k}}\text{Im}[G_{\text{lat}}(\omega,\mathbf{k})]_{\alpha,\alpha}$ with $G_{\text{lat}}(\omega,\mathbf{k}) = \left((\omega + \mu + i\delta)\mathbf{1} - H_0^d(\mathbf{k}) - \tilde{\Sigma}^{\text{DMFT}}(\omega)\right)^{-1}$ at $\tilde{U}_{dd} = 5\,\text{eV}$, $T \approx 150\,\text{K}$. The (light) purple line shows the spectral function (error bar) of $d_{x^2-y^2}$. A quasiparticle peak is observed with all doping levels. (b): $k_F$ of the nodal point compared to experiment. (c): $d$-model uniform spin susceptibility vs. Knight shift.

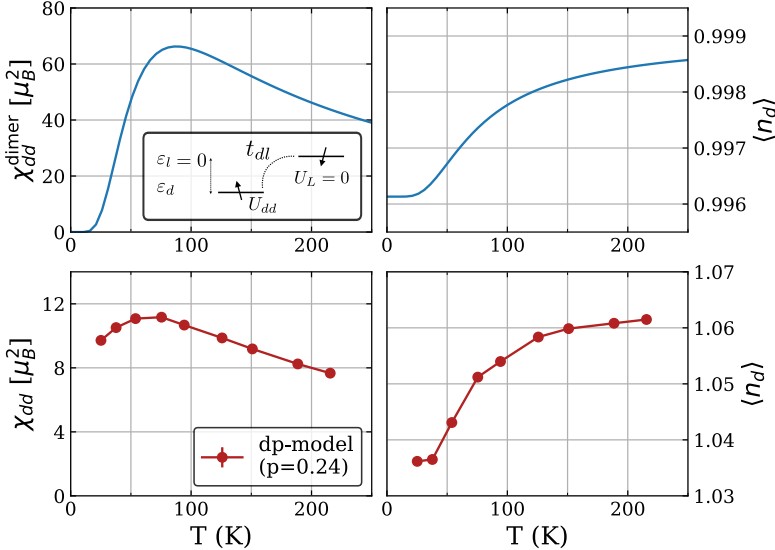

Figure 6: Static uniform magnetic susceptibility calculated in DMFT in the high doping limit $p = 0.24$ (lower panels) compared to an asymmetric dimer model (upper panels): Magnetic susceptibility $\chi_{dd}$ and density $n_d$ show the same trend as a function of temperature in both models. The unscreened moment of the dimer leads to a susceptibility which is about one order of magnitude larger than in DMFT.

Table 1: Eigenvalues and eigenvectors of the ground state and first excited state, respectively, of an asymmetric dimer. The notation $|\downarrow,\uparrow\rangle$ indicates spin down on the d-site and spin up on the ligand. The singlet state is the ground state, which causes decreasing susceptibility at lower temperatures.

| energy (eV) | eigenvector |
| --- | --- |
| 0 | $0.7|\uparrow,\downarrow\rangle - 0.7|\downarrow,\uparrow\rangle + 0.02|\uparrow\downarrow,0\rangle + 0.07|0,\uparrow\downarrow\rangle$ |
| $\approx 0.02$ | $1/\sqrt{2}(|\uparrow,\downarrow\rangle + |\downarrow,\uparrow\rangle)), |\uparrow,\uparrow\rangle, |\downarrow,\downarrow\rangle$ |

Due to the limitations of our toy model, which completely neglects finite bandwidth effects and is fixed to the high doping limit, we cannot compare the plots on a quantitative scale. While we refrain from over-complicating the toy model by fitting its parameters to the DMFT result, we fix the d-orbital parameter but rescale the ligand parameter $t_{dl} = 0.1\,\text{eV}$ to fit the effect of the hybridization function and to obtain a temperature scale that coincides with the material realistic DMFT calculation. The data shown in Fig. 6 shows a qualitative agreement between the behavior of DMFT and the asymmetric dimer model for both the $d_{x^2-y^2}$ occupation and downturn of the uniform susceptibility (It should be noted that due to the much reduced screening of the impurity moment, the susceptibility in the dimer is about ten times larger than the DMFT result).

The downturn of $\chi_{dd}$ at low-temperature regions in the dimer model can be traced back to the increased dominance of its singlet ground state (i.e., the local limit of the Zhang-Rice singlet). Importantly, the asymmetry of the dimer between copper $3d$ orbital and oxygen $2p$ ligand leads to a non-negligible temperature-dependent change of the $d_{x^2-y^2}$ occupation. Such a change would be completely absent in a symmetric dimer model which would be the corresponding toy-model for the single-band Hubbard model.

## C   Impurity susceptibility

In Fig. 7 we show the impurity susceptibility $\chi_{dd}^{\text{imp}} = \int_0^\beta d\tau \left\langle \hat{S}_z^{\text{imp}}(\tau), \hat{S}_z^{\text{imp}}(0) \right\rangle$ as a function of temperature and doping. Comparison to the uniform response $\chi_{dd}(\mathbf{q}=\mathbf{0})$ reveals an inverted doping trend for $\chi_{dd}^{\text{imp}}$, a behavior also reported in previous studies on the periodic Anderson model [50]. The decrease of $\chi_{dd}^{\text{imp}}$ as a function of doping indicates a lowering of atomically localized moments on the copper site and highlights the increasing importance of fluctuations around the Fermi level in determining the *uniform* magnetic response.

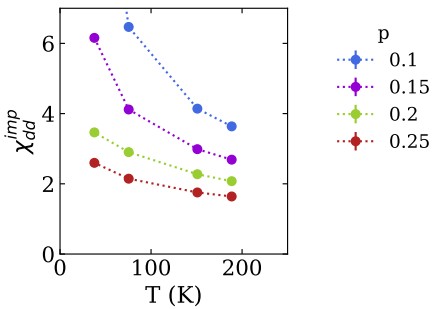

Figure 7: Impurity susceptibility $\chi_{dd}^{\text{imp}}$ of the auxiliary single impurity Anderson model as a function of temperature for different doping levels.



SciPost Phys. **18**, 145 (2025)

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
