# Peer review of "Single-particle spectra and magnetic susceptibility in the Emery model: a dynamical mean-field perspective"

_SciPost Physics, doi:SciPost Phys. 18, 145 (2025)_

## Round 2 · Referee Report · Anonymous (Referee 1) · 2024-10-29

Strengths

1 - Clear Presentation 2- The methodology is well established with its pros and cons. The manuscript leverages on this point 3- Solid information and simple message about a very important topic

Weaknesses

1- The connection with similar models (periodic Anderson model) is not discussed

Report

The manuscript presents a theoretical analysis of the Emery model for high-temperature superconductors using single-site Dynamical Mean-Field Theory (DMFT). DMFT only introduces local self-energy corrections, hence it is usually discarded as a method to study models for high-temperature superconductors, in which non-local correlations are universally considered fundamental. Yet the present work shows that, in contrast with the single-band Hubbard model, the three-orbital Emery model shows important non-local correlations owing to the multi-orbital character. This point is well discussed in the manuscript in general terms and its consequences are explored. In particular, the authors find that the q=0 uniform magnetic susceptibility $\chi_{dd}$ is remarkably different from the single-band Hubbard model (shown in the appendix A) and, remarkably, it reproduces some trends of the experimental data in cuprates, reproducing in particular the qualitative behavior in the pseudo gap region. This is particularly surprising and intriguing, as the DMFT single-particle spectra fail, as expected, to reproduce the pseudo gap. The authors discuss some arguments behind this seeming discrepancy (lack of feedback on single-particle properties of the behavior of the two-particle response given by the spin susceptibility) and the perspective to overcome it by using D$\Gamma$A.

I find this result very intriguing and I support the publication of the manuscript in SciPost Physics. The use of the very well established single-site DMFT and the straightforward calculation of the uniform susceptibility computing the response to a small static magnetic field make the conclusions very solid and they identify an non trivial effect. I am also intrigued by the simple two-site model described in the appendix B. Indeed this oversimplified model captures the main trend of the DMFT calculation. This success also suggests that the results discussed here can be related to those for the periodic Anderson model (PAM). Indeed in Phys. Rev. B 85, 235110 (2012) the PAM has been solved within DMFT, finding the formation of Zhang-Rice singlets similar to those reported here. In that paper the authors compute the local (impurity) spin susceptibility which however has a Curie-like behavior. So, I am wondering if this is the case also for the Emery model, while the non-Curie behavior is found in the q=0 response, which is notoriously different from the local counterpart in DMFT.

As I detail below, this comparison with the local susceptibility (and possibly with the PAM) could help to strengthen the manuscript.

Another suggestion I have is to consider a title/presentation which makes it more explicit that the two-particle observables mentioned in the title coincide with the uniform susceptibility. The current version of the title gives the incorrect impression that the manuscript would show dynamical two-particle observables which have been computed in a number of papers from the Vienna group and coworkers.

I detail below also other points that I would like the authors to address in order to meet the criteria for publication. A revised version meeting all my points should be published in SciPost Physics.

Requested changes

1- I would eliminate the sentence about static mean-field reproducing the undoped system. While an AFM is typically found, the mean-field description is quite far from the actual situation and from DMFT.

2- At line 10-11 of page 2, I would stress more that the drop of the Knight shift is generally interpreted as resulting from the reduction of spectral weight, while the present calculation finds it without a spectral pseudo gap.

3- The authors should discuss what is the choice of the double counting used here and whether also the interactions are extracted from ab-initio calculations. I would also mention the value of the d-p repulsion even if this is neglected, since this is probably the most important simplification introduced in the model

4- The authors write that the susceptibility is computed for a value of h $\simeq$ 5 meV. Why $\simeq$ and not just equal? Do I understand correctly that the authors simply use a linear approximation connecting zero field with this value of h? Did they test if this value is small enough for the present calculations? (observing that no change is introduced by reducing h)

5- minor: I think that the Clogstone-Jaccarino plot should be explained to the general readership

6- As mentioned above, I would suggest a title/presentation making more explicit that the two-particle observables mentioned in the title coincide with the uniform susceptibility.

7- I would add if possible one example of the local spin susceptibility to compare with the q=0 response. This can also be connected to a comparison with the periodic Anderson model.

8- In Fig. 2 the x and y range of the panels (a) showing the self-energy are cut in a rather extreme way. I would extend them to highlight better the evolution of the real and imaginary part of $\Sigma_{dd}$.

Recommendation

Publish (surpasses expectations and criteria for this Journal; among top 10%)

  • validity: high
  • significance: high
  • originality: high
  • clarity: top
  • formatting: excellent
  • grammar: excellent

Author:  Philipp Hansmann  on 2025-03-03  [id 5256]

(in reply to Report 1 on 2024-10-29)
Category:
answer to question

We provide a detailed reply to all comments in the attached pdf file.

Attachment:

response_to_reviewer1.pdf

---

## Round 2 · Referee Report · Anonymous (Referee 2) · 2024-10-30

Report

The authors address the physics of the three-band Emery model within the single-site DMFT approximation. The calculations are performed for a set of parameters, which has been recently suggested as material realistic for the cuprate superconductors. As the authors state themselves, single-site DMFT comes with limitations, foremost the failure of momentum resolving spectral functions crucially required to unravel pseudogap physics. However, this work can lay a ground on which other more advanced techniques can be rooted and should be seen as a first step towards more elaborate studies. T

Within the limitations of the single-site DMFT approximation, the calculations appear to be competently carried out, and the results are presented concisely. I would nevertheless suggest several improvements to the readability, as in the current form, the manuscript only seems accessible to people at the close intersection between the DMFT community and the cuprate community.

1) In Fig. 1, the color code of the band structure is referred to as "orbital character". As far as I can see, no definition in the text is provided and should be added. 2) The authors repeatetly mention the Zhang-Rice singlet state without an explanation. The readability of the manuscript can be improved by incorporating a concise explanation of the Zhang-Rice singlet. Moreover, in Fig. 2 it should be explained why the zero energy peak corresponds to the Zhang Rice singlet. This is currently not explained well in the manuscript. 3) The DMFT method is explained only very briefly in section 2. This should be expanded and a discussion of strengths and limitations should be included. 4) The shown spectral functions show rather fine features, which is remarkable as they apparently have been obtained from analytical continuation. The manuscript should explain in further detail, to which accuracy the Matsubara Greens functions are calculated and how this leads to such precise predictions on the real-frequency Greens functions. 5) In general, all energies are reported in electronvolt. However, to broaden the scope for other computational studies it would not hurt to also include dimensionless units, e.g. in units of some hopping in the three band model. 6) In Fig 3., the authors mention that the self-energy shows Fermi liquid behavior. The authors should specify explicitly what is meant by this and how this can be seen. 7) Similarly, the authors repeatedly mention non-Curie like behavior of the magnetic susceptibility. This should be made more specific, e.g. by saying that no divergence but a maximum is observed. 8) The authors show a Clogston-Jaccarino plot but do not explain what this means. This plot should not be considered widespread knowledge and a short explanation would be helpful. 9) In the conclusion, the authors mention that a single band Hubbard model seems to give precise predictions for the nickelates. This statement should be weakened, as much of the community seems to agree that multi-orbital effects in the nickelates are of crucial importance.

I consider the present manuscript a valuable contribution to the literature and am in favor of publication. However, I am sceptical about whether the paper meets the bar of being among the top 20 percent of PRB publications required for SciPost Physics. It is unclear which of the findings will survive a treatment with more accurate approximations beyond single-site DMFT. Even if there is better agreement with cuprates than in the single band case, it is not excluded that this is a coincidence, even though it might be an excellent hint already. I would consider a study using cluster DMFT revealing the pseudogap behavior to meet this bar, but a single-site DMFT calcution might just be too crude to induce confidence the the observed behavior is a trustworthy feature of the genuine two-dimensional Emery model. I would, therefore, rather suggest publishing the manuscript in the SciPost Core series, upon my requested changes are implemented.

Recommendation

Accept in alternative Journal (see Report)

  • validity: top
  • significance: good
  • originality: ok
  • clarity: top
  • formatting: excellent
  • grammar: excellent

Author:  Philipp Hansmann  on 2025-03-03  [id 5257]

(in reply to Report 2 on 2024-10-30)

We provide detailed answers to all comments/suggestions in the attached pdf file.

Attachment:

response_to_reviewer2.pdf

---

## Round 2 · Referee Report · Anonymous (Referee 3) · 2024-11-11

Report

The authors present the solution (one-particle spectral function, self-energy and uniform susceptibility) of the three-band Emery model within the single-site DMFT approximation. In this model the electrons in the oxygen p-orbitals are treated as non-interacting and only the d-d onsite repulsion is taken into account. The DMFT approximates the solution by neglecting dynamical nonlocal correlations, which in this case means correlations beyond the unit cell. The parameters of the Emery model are adapted to describe the cuprate superconductors as in earlier cluster DMFT studies of Refs. 34 and 35. More precisely, one set of parameters is used. The one-particle spectra are discussed and compared directly with the one-band Hubbard model (also computed by the authors and placed in the Appendix) and indirectly with earlier cluster studies. The lack of momentum dependence in the self-energy and hence absence of the pseudogap is addressed and at the one-particle level the deficiencies (and also strengths) of DMFT carefully addressed. The authors then present two-particle calculations on the example of static uniform susceptibility, computed by performing computations at finite magnetic field. The numerical results are then compared to the experimental measurements of the Knight shift and (in the Appendix) to the one-band Hubbard model in DMFT as well as to the asymmetric dimer toy model. They find that the asymmetric dimer represents the qualitative temperature dependence of the susceptibility much better than the DMFT calculation for the one-band Hubbard model, which the authors assign to the singlet formation captured well by the dimer or by the Emery model (even in DMFT), but apparently not in the DMFT for the one-band Hubbard model. The single-site DMFT calculations are thought of as a starting point for a future D$\Gamma$A study of the Emery model.

I find already the DMFT results interesting and certainly deserving publication. What I miss in the current version of the manuscript is a thorough discussion of two-particle properties as captured or not captured by DMFT. It is not clear to me why the uniform susceptibility is the correct quantity to represent the Knight shift in place of the local susceptibility. The authors should explain if this is related to the fact that the computational method used gives better results for uniform susceptibility, because it can be computed directly as a derivative and not from the correlation function (which I suspect to be the case), or to the actual way the experiments are performed. In case the authors address this point in detail, I recommend the paper for publication.

Minor points:
(1) I would appreciate a justification of the choice of the value of the onsite repulsion. Do the authors expect the results to depend on the precise value?
(2) Optional: A third plot in Fig. 6 with the d-only spin susceptibility (if data are available for the relevant temperatures) could make the argumentation stronger.
(3) The Clogston–Jaccarino plot should be explained.

Recommendation

Ask for minor revision

  • validity: -
  • significance: -
  • originality: -
  • clarity: -
  • formatting: -
  • grammar: -

Author:  Philipp Hansmann  on 2025-03-03  [id 5258]

(in reply to Report 3 on 2024-11-11)

We provide detailed answers to all comments/suggestions in the attached pdf file.

Attachment:

response_to_reviewer3.pdf

---

## Round 3 · Referee Report · Anonymous (Referee 3) · 2025-3-24

Report

In response to the Referees, the authors decided to include also 2x1 cluster DMFT calculations that are consistent with the DMFT calculations for the uniform susceptibility at least for higher doping levels. As I already wrote in the first report, I find the DMFT results interesting and deserving publication in SciPost. The 2x1 calculations do not in the end bring much, because it is anyway not clear what happens when longer range correlations are fully taken into account. As I understand, this is planned for a future study.
The authors addressed satisfactorily all my minor points. What I still miss is a deeper discussion on what is captured by which susceptibility in DMFT. The authors do show the DMFT impurity susceptibility in the appendix now and indicate in the main text why the uniform susceptibility captures the correct behavior even within DMFT (similarly to the case of the periodic Anderson model, results for which the authors cite). I would have wished however for a more extensive discussion. I leave it to the authors as optional though. In my opinion, the revised paper fulfills the criteria to be published in SciPost.

Remark: In the caption of Fig. 2 should be UHB for upper Hubbard band instead of LHU.

Recommendation

Publish (easily meets expectations and criteria for this Journal; among top 50%)

---

## Round 3 · Referee Report · Anonymous (Referee 2) · 2025-3-27

Report

The authors have addressed my previous concerns and issues comprehensively and the resubmitted manuscript has improved significantly. Especially the addition of further CDMFT data now puts the study on a firmer basis. Hence, I can now fully recommend publication in SciPost Physics.

Recommendation

Publish (meets expectations and criteria for this Journal)

---

## Round 3 · Author Response

Dear Editor,

We sincerely thank the Referees for their thorough reviews and their overall positive feedback. We appreciate the time and effort taken to provide constructive comments, which have helped us to improve the manuscript significantly. We now submitted our detailed responses to each referee and hereby a revised version of the manuscript with the requested changes to be further considered for publication in SciPost Physics. The most important addition to our manuscript are new CDMFT results for the uniform magnetic susceptibility which support our previous claims.

We hope that our replies and the additional calculations are appreciated and that our revised manuscript fulfils now the criteria for publication in SciPost Physics.

Best regards,
Yi-Ting Tseng, Mario O. Malcoms, Henri Menke, Marcel Klett, Thomas Schäfer, and Philipp Hansmann

---

## Round 3 · List of Changes

• Following Referee 2, we performed CDMFT calculations for the uniform susceptibility in order to strengthen our conclusions. These new data are now shown in the results section.
  • We have emphasised that the non-Curie behaviour of the susceptibility does not require the opening of a pseudogap in the spectral function.
  • We have added remarks about the choice of the interaction parameters and the neglect of Upp and Upd.
  • We have added a more comprehensive explanation of the Clogston-Jaccarion plot.
  • Following Referee 1 we changed the title to be more specific about the presented results.
  • We added a comment and a plot in the appendix about the relation of the Emery model to the periodic Anderson model and the different doping trends of the uniform to the impurity susceptibility.
  • We added insets in Fig.2 to show the behaviour of the self-energy also on a wider scale.
  • We added comments about the definition of the color coding in Fig.1.
  • We provide additional references for the notion of the Zhang-Rice singlet.
  • We extended the comments about the limitations of DMFT and extended a bit the description of the analytical continuation of the self-energy.
  • We state now more clearly what “Fermi-Liquid behaviour” means in the context of the self-energy.
  • We softened the statement about the difference between minimal models debated for cuprate- and nickelate- superconductors.
  • Further we have made several small edits and corrections of typos that were indicated/found by the referees and ourselves.

---

## Editorial Decision

published